# Mesoscale structure development reveals when a silkworm silk is spun

Quan Wan[1], Mei Yang[1], Jiaqi Hu[1], Fang Lei[1], Yajun Shuai [ID] [1], Jie Wang[1], Chris Holland [ID] [2✉], Cornelia Rodenburg [ID] [2✉] & Mingying Yang [ID] [1✉]

Silk fibre mechanical properties are attributed to the development of a multi-scale hierarchical structure during spinning. By careful ex vivo processing of a *B. mori* silkworm silk solution we arrest the spinning process, freezing-in mesoscale structures corresponding to three distinctive structure development stages; gelation, fibrilization and the consolidation phase identified in this work, a process highlighted by the emergence and extinction of 'water pockets'. These transient water pockets are a manifestation of the interplay between protein dehydration, phase separation and nanofibril assembly, with their removal due to nanofibril coalescence during consolidation. We modeled and validated how post-draw improves mechanical properties and refines a silk's hierarchical structure as a result of consolidation. These insights enable a better understanding of the sequence of events that occur during spinning, ultimately leading us to propose a robust definition of when a silkworm silk is actually 'spun'.

[1] College of Animal Science, Zhejiang University, Hangzhou, China. [2] Department of Material Science and Engineering, University of Sheffield, Sheffield, UK. ✉email: christopher.holland@sheffield.ac.uk; c.rodenburg@sheffield.ac.uk; yangm@zju.edu.cn

Silks are a group of proteins that are spun into fibres for use outside of the body. The process of spinning, i.e. the instigation and control of protein self-assembly from an aqueous feedstock into a solid fibre, has been of significant scientific and technological interest for over a century[1–3]. After considerable global effort, it is now possible to approach the remarkable properties of the natural fibres through a range of artificial means, although moving beyond a technological perspective on silk, there are still fundamental gaps in our knowledge surrounding structure development in the natural system.

In order to address these gaps, we previously proposed a comparative framework for silk fibre production[4,5]. This discussed that while there is good experimental and modelling evidence for events that relate to protein interactions at the nanoscale[6–8], such as those probing the facilitative roles of the N and C termini[9–12], at the mesoscale, structures generated by inter-protein interactions are surprisingly challenging to characterise and subsequently model with high resolution[13–15]. These structures range from micelles or granules[16,17], optical textures[18,19], nanofibrils[20,21] and the peculiar presence of voids or canaliculi[22,23]. Hence, while recent progress has been made in recombinant spider silk systems[9] with an increasing body of evidence, it is clear that there exists a disconnect between the diversity of mesoscale features present in silks produced from a variety of animals. This demands more systematic attention and explanation, in order for the field to reach a consensus surrounding a general picture of silk fibre development at this length scale.

The focus of this study is to ascertain the mechanistic origin of these features and to attempt to place them within a continuum of structure development as silk undergoes a flow-induced phase transition during spinning. Ideally, this would be probed in vivo, and while this is becoming a reality for nanoscale structures via spectroscopy[6,24], at the mesoscale there are currently significant practical and technological challenges that make this an impossibility. Hence previously this has been approached bottom-up, by looking at the structures formed by an aqueous silk feedstock as it subjected to a range of thermal, chemical and mechanical stress fields which serve to initiate protein self-assembly and denaturation[25,26]. An alternative approach is top-down, reverse engineering features present in the spun fibre, usually as a result of different spinning conditions[27]. Taken together, important insights have been gained via individual techniques but issues surrounding inter and intra-sample variability mean that, while some features have been observed and studied in great detail, a cohesive story has remained elusive until now.

In this work, we propose that combining a top-down and bottom-up approach may demonstrate the diversity of features required to fill in the gaps in our knowledge. As a model system, we have opted to investigate silk from the mulberry silkworm *Bombyx mori*, due to its prevalence and accessibility, with the aim that such an approach could be applied to the more specialised spider, insect and arthropod silks in general in the future.

## Result and discussion

In order to investigate structure transition in silk spinning, we replicate the natural process by subjecting a regenerated *B. mori* silk feedstock to an extensional flow field in order to partially draw fibres[14,28–30] and then viewing the structures formed as the silk proteins self-assemble using low-voltage SEM (SE and BSE modes) on cryo-sectioned uncoated samples (Fig. 1).

Figure 1, illustrates the rich diversity of structures present through this arrested development approach. This cross-section of a partially spun regenerated silk fibre (Fig. 1a) clearly contains features that have been seen in natural silkworm silk fibres that

are an order of magnitude smaller in diameter (Fig. 1b–e), such as fibrillar structures on the surface[13] and the presence of nanoscale voids which we propose are more appropriately referred to as 'pockets'[31]. Notably to our knowledge these pockets are not readily observed in *B. mori* silks, but are in wild silkworm silks, suggesting that they may be under different selective pressures (artificial vs. natural). The natural variability (in terms of shape and scale) of pockets observed in naturally spun silk in Fig. 1 suggests that their formation, transitory nature and persistence in the fibre is likely to be a manifestation of both the composition of the fibre and the spinning process[32,33]. However, while this sample is substantially larger than the natural fibre (and possesses a breaking stress of only 200 MPa (see supplementary information table 1)), the raised centre and large porous regions are suggestive of a non-Newtonian flow profile and the stress-induced phase separation proposed by Knight[34] and seen in rheological studies since the work of Iizuka in the 1960s[35]. In particular, we consider these features as indicative of the progress of fibre formation and subsequently sectioned a sample as it was drawn from gel to fibre under controlled conditions (2 mm/min) (Fig. 2).

At the microscopic scale, it is clear from the composite bire-fringence image of a dried sample in Fig. 2a that orientation develops as the sample is drawn from the gel (0.002) to the fibre, eventually reaching similar values seen in natural silks (~0.02)[36] suggesting similar extensional flow processing mechanisms[37,38]. Sectioning and SEM imaging at points along with this sample links changes in birefringence to mesoscale structure development (Fig. 2b–f). Sectioning in the gel-like portion reveals a fracture surface that is largely amorphous, with deformation occurring in multiple randomly distributed locations, suggesting a plastic mode of failure rather than a fibrillar mode (Fig. 2b)[39,40].

The point at which the birefringence starts to increase significantly, SEM reveals a fibril-like pattern on the outer edge of the fibre (Fig. 2c and for more detail see Supplementary Information Fig. 2). Towards the centre of the fibre, the fracture surface becomes smoother and raised, which suggests a cross-sectional gradient in the mechanical properties, caused by the internal structure deforming in line with pultrusion orientation[30,41]. Upon magnification, this raised section also reveals the presence of pockets, which are more concentrated at the centre of the fibre (Fig. 2d), akin to the distribution seen in Fig. 1a. We attribute the non-Newtonian properties of the silk solution and the extensional flow generated by drawing to be responsible for the fact that pockets appear to form at the centre of the fibre, as this section encounters the highest degree of deformation during spinning[42].

As the birefringence increases and the fibre appears more solid, we observe the pockets starting to radiate and spread out across the entire surface of the fibre, with each pocket surrounded by needle-like 50–70 nm diameter nanostructures (Fig. 2e). However, of key importance is that where the sample birefringence reaches its maximum, these pocket and needle-like nanostructures are no longer observed and the fibre surface becomes smoother and less feature-rich (Fig. 2f).

The appearance and subsequent disappearance of these pockets provide a remarkable insight into silks' solidification and may be considered a hallmark of the drawing process, as they do not exist in the gel and seldom in the fibre. Given that there is a significant change in silk hydration during both natural spinning[43] and in our system (from ~72% to ~7% of total weight) we assume that these pockets contain water released from the silk proteins as they denature (i.e. dehydrate and change conformation)[44] and aggregate into nanofibrils in response to the applied extensional stress field[4]. As this drawing continues, we propose the nanofibrils of silk consolidate into mesoscale bundles and the water is expelled

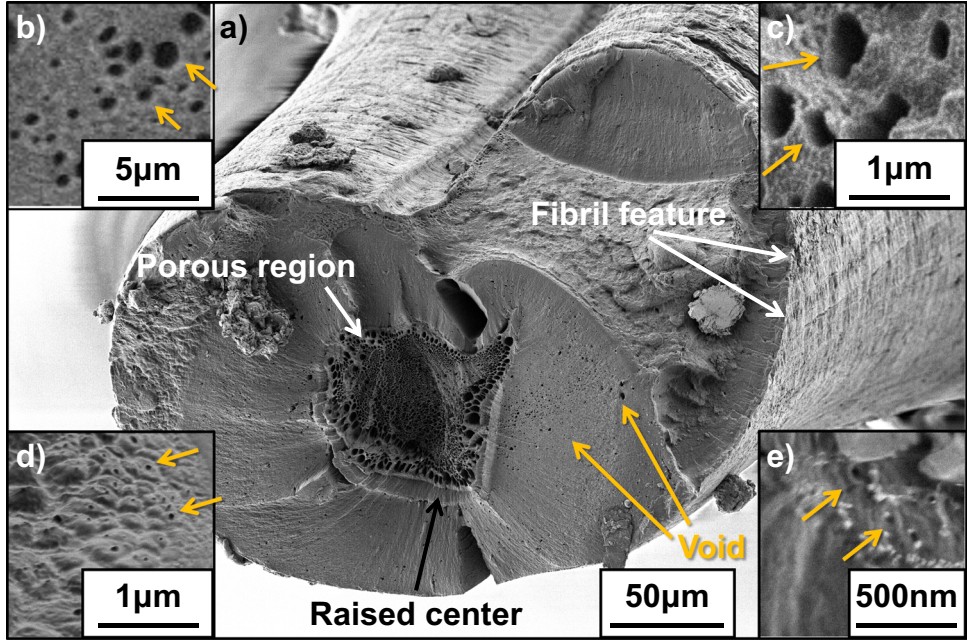

**Fig. 1 A variety of porous region related mesoscale structures are observed in natural and artifical silk fibres.** Formation of mesoscale structures in a partially spun regenerated silk fibre, featuring a porous region in a raised central area and sub-micrometre scale voids (**a**). Void structures are identified by orange arrows. As an example of the generality of our observations, a magnified area of this sample (**b**) is compared with similar porous regions and nanofibrillar structures present in cryo-cross sections of native *Antheraea mylitta* (**c**) and the outer edge (**d**) and central region (**e**) of *Antheraea pernyi* silk. While indicative, it is worth noting the diversity, scale and distribution of these mesoscale structures are likely to be specific to each species and spinning conditions, much like fibre mechanical property variation on an intra and interspecific level.

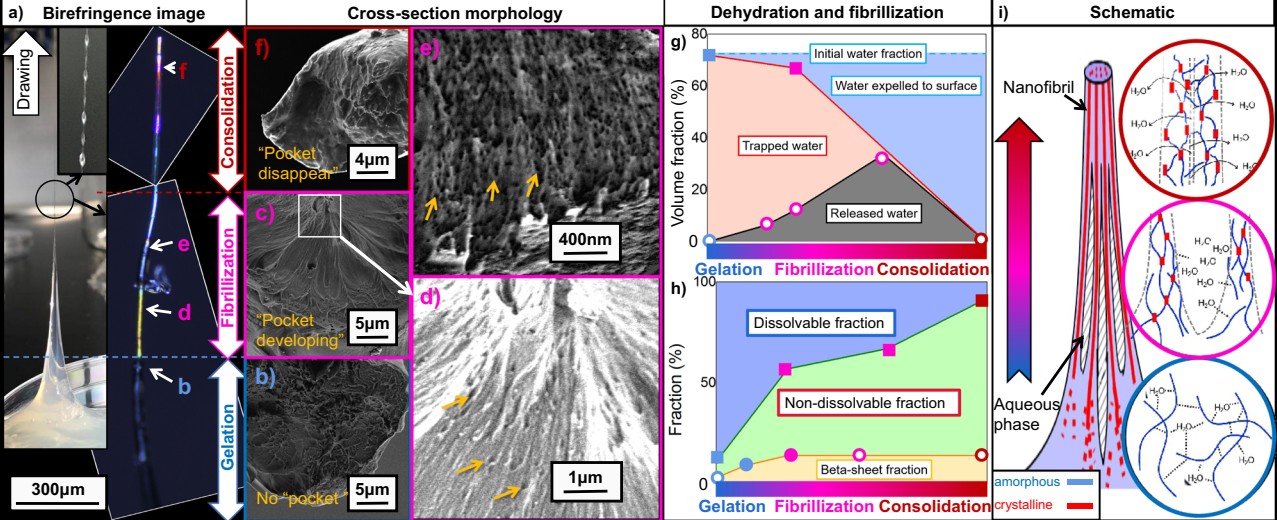

**Fig. 2 The structural development in the formation of a silk fibre is demonstrated in this figure.** A drawn silk feedstock and subsequent imaging of the last few millimetres under cross-polars (**a**), note the gradual increase in birefringence as structure develops. Inset: the water droplets expelled to the fibre surface as it solidifies. SEM images of fractured cross-sections of each segment **b–f** of the drawn fibre are shown in the image (**b–f**), **c** the inset region indicated by the white box is magnified and contrast-enhanced to show the development of water pockets (yellow arrows), which become more common as nanofibrils are assembled (**e**), eventually leaving only the ~30 nm diameter 'needle-like' structures are left solid besides water pockets, **f** these pockets subsequently disappear and needle-like structures fuse into a dense fibre. Dehydration-related structural changes are presented in **g** where the grey region represents the volume fraction occupied by the water pockets, the red region represents residual water within protein domains (calculated by initial solution water fraction minus phase-separated and expelled water) and the blue region represents water expelled to the sample surface (measured by careful mass comparison before and after removal of surface water droplets). The corresponding increase of non-dissolvable protein is shown in **h**, where the yellow region represents the beta-sheet crystal fraction, the green region represents the non-dissolvable protein fraction and the blue region represents the dissolvable protein fraction. In **g** and **h** data points marked by hollow circles were derived from SEM images, data points shown as solid circles were obtained by Raman spectroscopy, and data shown as solid squares were calculated from measured mass differences. A schematic of the complete process is shown in **i** where blue chains represent amorphous fibroin and red blocks represent beta-sheet crystals.

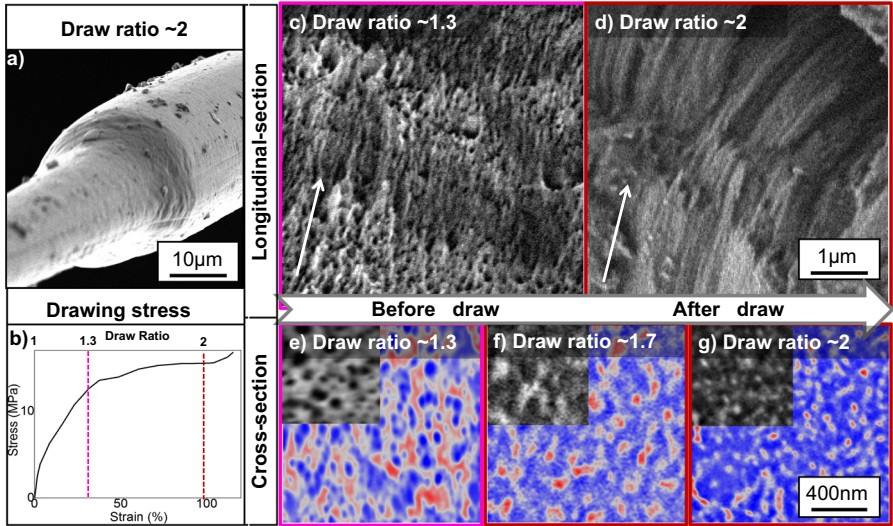

**Fig. 3 A fibrillization stage sample as shown in Fig. 2 was further processed to consolidate in this figure.** This sample was further drawdown under wet conditions until it entered the consolidation stage (**a**). The stress applied in this drawdown process at a constant draw speed of 5 mm/min is shown in plot (**b**). The longitudinal section (**c**, **d**) and cross-section (**e**–**g**) of fibres subjected to differing levels of draw are compared by SEM imaging to show structure development, the fibre longitudinal axis is marked by a white arrow in **c**, **d**. SEM images are taken in backscattered mode, with contrast corresponding to local density[64]. Red in the colour coded images represent areas of higher density and blue represent areas of lower density alongside pockets.

to the surface, which supports previous evidence of nanoscale dendritic 'riverbed' patterns[45] in sectioned spun *B. mori* silk fibres where pockets are not seen. Hence the structure development seen in Fig. 2, occurs alongside a major dehydration process, as depicted in the schematic in (Fig. 2i) which we define as three stages (1) gelation, (2) fibrilization and (3) consolidation.

From our description above it follows that the pocket volume (as determined from SEM images) corresponds to phase-separated water. This is supported by evidence that upon exiting the spinneret natural silk fibres retain a considerable amount of water (~70%) which is mostly lost while it is drawn through the air[43,46,47]. Hence we can plot the dehydration process versus structure development as shown in Fig. 2g. This allows us to define water as being either trapped in the protein phase, phase-separated into water pockets or expelled to the fibre surface. To account for this we suggest there exists an equilibrium between water in the protein phase, the pockets, and water being lost to the surface via diffusion which is driven towards further dehydration through the extension of the fibre (which has previously shown to occur in an amorphous silk protein phase[22]). This balance would also be influenced through the environmental humidity, as this has been shown to impact final fibre properties and more recently fibre formation through extension[30,48,49].

Our data indicate that as the fibre is formed, nanofibril consolidation as a result of accumulated extensional flow (physical stretching) combined with volume constancy of the solid portion of the fibre, forces water to be 'squeezed out' of the pockets and left to diffuse towards the surface through a combination of capillarity, osmotic pressure and matrix interaction as the silk proteins become denatured and relatively more hydrophobic. However, the time and length scales involved in this process are crucial from a mass transfer perspective, as fibre dehydration cannot be completed if the resulting diffusion path is too long during the time of fibre formation (i.e. the fibre is too thick or spun too fast). Figure 1 shows such a situation.

As a means to confirm that the loss of water is associated directly with silk protein dehydration and solidification akin to natural spinning, we determined the β-sheet secondary structure fraction of each segment (from backscattered SEM images and Raman spectroscopy, see methods) and the propensity of fibres

experiencing the same processing conditions to dissolve in water (Fig. 2h). The results indicate that as drawing increases, so does the non-dissolvable fibre and β-sheet fraction. However, the non-dissolvable fraction continues to increase after the point at which the β-sheet fraction plateaus. This implies that even after the structural conversion of the silk proteins is complete, the fibre is not fully solidified, as some of it can still dissolve (presumably the non-consolidated amorphous regions). In addition, the Raman signal (as shown in supplementary information) corresponds to such dehydration is reported to be essential for the stability and folding behaviour of the amorphous fibroin chain. The report demonstrated interplay of N- and C-terminus in fibroin H-chain caused a series of behaviour in fibroin transformation[50]. The spacial factor of peculiar peptide versus chain terminus is proposed taking part in molecular-level mechanism[51,52], this theory suggests the possibility of fibroin chain interacting with neighbour nano-structure in dehydration.

The identification of three distinct stages of mesoscale structure development in silk surrounding gelation, the creation of nanofibrils and their subsequent consolidation, provides us with a platform to probe how processing relates to structure development and mechanical performance of the fibre. The fact that fibre mechanical properties can be controlled through spinning rate and post draw is well-established[53], and is often related to the increase in crystallinity or orientation of the fibre[27]. However, questions remain around whether the hierarchical structures seen in fibres developed during spinning can be replicated simply by post drawing fibres alone.

To address this, we compared samples that initially exhibited intermediate birefringence (~0.0067, which is comparable to segment Fig. 2d), by subjecting one of them to further extension at 5 mm/s until a ~2:1 draw ratio was reached (Fig. 3a, b) Post drawing saw a marked change in the mesoscale structures present in the fibres, akin to those seen in Fig. 2. Prior to the drawing, SEM revealed the presence of a distributed network of water pockets in the fibre (Fig. 3c, e). After post drawing, these water pockets disappeared and fibrillar structures were visible (Fig. 3d)[13]. Moreover, in cross-section, a more clearly defined phase transition was observed with 40 nm crystalline ordered structures being surrounded by disordered regions (Fig. 3f, g) Note that beyond a ~2 draw ratio the fibre began to strain harden,

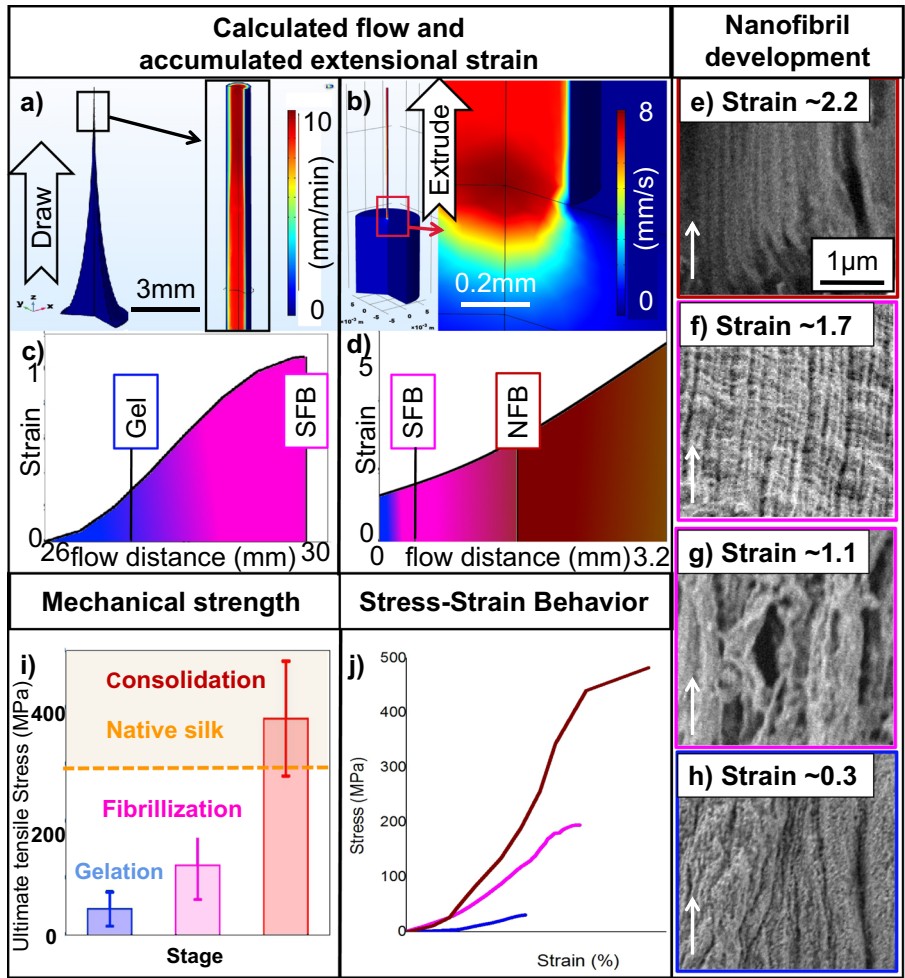

**Fig. 4 The development of silk strucutre and property is correlated with extensional strain accumulation.** The flow field calculated (**a**) for the sample before drawdown in Fig. 2 is shown along with the accumulation of extensional strain (**c**) versus the distance of sample extensional flow. The same calculation of an extruded sample is shown in **b** and **d**. The colour gradient from blue to red represents a low to high flowing rate. Note the simulation is built using an axial symmetry approximation and is not the accurate value due to limitations of the calculation. The samples showing slippery fibril bundle (SFB) structure and non-slippery fibril bundle (NFB) are marked in plots **c** and **d**. The development of a nanofibrillar structure with respect to the strain accumulation is shown in longitudinal SEM images (**e**–**h**) with the extensional flow direction marked by white arrows. The average sample mechanical strength of each stage is shown in plot (**i**). A representative stress-strain curve of each stage sample is shown in plot (**j**). The error bar represents the standard deviation for the tensile strength of each stage, which is ±21.3, ±44 and ±74.7 MPa, respectively, $n = 5$ for the gelation stage and $n = 6$ for the other two stages.

implying complete solidification and hence we assume no more mesoscale structure development takes place.

This implies that post draw can be applied to progress, not just nanoscale, but also mesoscale structure development in partially spun fibres. Furthermore, we find that if processed carefully and drawn to the point at which mesoscale structure development is complete but not beyond, these fibres exhibit comparable mechanical strength to natural fibres (as shown in Supplementary Information). This indicates artificial silk systems may be able to achieve natural silk structures and properties with carefully and appropriately applied strain, even when they are not spun in a biomimetic manner. However, this relies on the fact that the feedstock proteins are not degraded to an extent that precludes their ability to undergo natural self-assembly pathways[54,55] (discussed further in Supplementary Information).

As these results suggest structure development through extension can occur during, or even after spinning, we addressed whether the process by which extension is applied has an impact on silk gelation, fibrilization and consolidation. This was achieved by comparing droplet spinning and draw (pultrusion) to extrusion via the amount of accumulated extensional strain achievable[42,56]. Through a combination of our modelling predictions and experimental preparation, it was possible to create pultruded samples with a calculated accumulated extensional strain of up to 1.1 and extruded samples to between 1.2 and 5.6. (Fig. 4a–d).

Exposing samples with varying accumulated extensional strains to water and imaging the portions that do not dissolve (Fig. 4e–h), we see a similar three-stage structure development to fibres that had undergone post draw (Fig. 2). At low levels of accumulated strain (<0.5), samples remained gel-like and displayed little-to-no structure development. As the level of accumulated strain increased, nanofibrils begin to assemble in the flow direction into structures akin to slippery fibril bundles[57]. As strain increases, the nanofibrils become more aligned and consolidated while remaining partially dissolvable (Fig. 4g) up to an extensional strain value of 1.7 (Fig. 4f). Beyond an accumulated extensional strain of ~2.2, the system is no longer dissolvable and presents a consolidated non-slippery fibril bundle (Fig. 4h). This gradual change in mesoscale properties is also

matched by a concurrent change in mechanical properties, with a consolidated fibre exhibiting a breaking strength similar to that of a native silkworm silk (Fig. 4i, j).

Combined, the findings in Figs. 3 and 4 suggest our silk protein system will progress into its final consolidation stage with ~2.2–2.4 accumulated extensional strain, regardless if it is via extrusion or drawing. This is close to the calculated value (~2.6) of natural silkworm silk spinning[42] and has the mechanical properties to match (Fig. 4i)[13,20]. Such similarity between our regenerated silk fibre and native silk fibres suggests that this consolidation stage identified in this work represents the end of the silk-spinning process, natural or artificial, with fibres that no longer show structure development may be considered fully 'spun'.

However, while we have gained important insights into silk self-assembly, this work represents the simplest case scenario for natural spinning. There is a huge degree of specialisation observed in silk producers, from internal drawdown to co-pultrusion, to highly complex post draw[41,58–61]. Hence our observations may be useful in investigating the role these more technologically advanced spinning processes may play in structure development; serving as a simple standard for degree of completion in silk spinning which directly relates to a common process variable, draw[2]. In the future we believe that such insights will help make it possible to directly and quantitatively link a top-down process variable such as draw (or extension) to the molecular-level energetic input into a silk feed-stock, allowing one to predict bottom-up hierarchical self-assembly and macroscale mechanical properties[2,15]. This will not only be of interest to the biological sciences, but also the biotechnologists and engineers seeking to develop artificial fibre processing and uses of silk and silk-like materials in biomaterial applications[62].

In conclusion, we demonstrated mesoscale structure development during silk protein spinning, a process fuelled through extensional flow, protein denaturation, aggregation and water removal. By defining three distinct phases of morphological and mechanical property development, we have been able to integrate previously disconnected pre and post spinning processing into a single continuous spectrum, which will not only help define when a silk is truly spun, but also act as a benchmarking system for bioinspired fibre production and future multi-modular modelling approaches[30,63].

## Methods

**Regenerated silk preparation.** *Bombyx mori* silk cocoons were cut into ~1 cm × 1 cm sized pieces and mildly degummed in a faintly boiling 1.25% aqueous solution of citric acid (Aladdin, 99.5%) (1:40 bath ratio) for 20 min, then rinsed thoroughly in deionized water. This degumming process was repeated twice to extract the majority of the sericin, and then the remainder was washed with hot deionized water before air-dried at 50 °C. The degumming process was controlled to optimise the solution viscosity of our regenerated silk protein for fibre drawing, more details are shown in the Supplementary Information. Degummed silk was dissolved in 9.3 M LiBr (Macklin, 99.9%) aqueous solution at 60 °C until no foam or suspended solids were present. The resulting solution was then dialysed against frequent changes of deionized water for 24 h, until Br+ ion cannot be detected by an AgNO₃ (Aladdin, 99.8%) titrant in the dialysis water.

Depending on the degree of degumming, the viscosity of this solution varies from a semi-solution-gel state to a viscous solution. However, the final concentration of the solution is always ~4% wt. Stock solutions were then further concentrated by air drying at room temperature to prepare different concentration systems. Multiple sets of solution were prepared and applied in our sample spinning. Each one is concentrated to ~27% wt concentration by air drying, the concentrating process took 6–12 h at <15 °C to avoid gelation. This concentrated solution is a slightly white, translucent solution gel.

The molecular mass of this protein solution is measured by the standard SDS-PAGE method. The protein sample is transferred into a 1 ml centrifugal tube with loading buffer and immersed into boiling water for 30 s, before injected onto 15% acrylamide gel. The running voltage is 120 V at the start and 140 V at the end, the current is 90 mA at the start and 30 mA at the end.

**Fibre drawing, extrusion spinning and strain calculation.** Concentrated silk protein solution-gels were placed onto a compression pattern of an Autograph universal tester (Shimadzu, Japan) equipped with a 5 N load cell and subjected to drawing at 2 mm/min until the drawing stress became unstable (more detail is discussed in supplementary information). The sample was then left under tension to dry in the air for 10 min after drawing ceased. Environmental humidity in our experiment is below 37%. The solidified segment at the end of the drawing process was collected for characterisation.

The fibrilization stage segment for post drawing was made by cutting off a dried drawn region exhibiting a birefringence below ~0.08 ($n_1$–$n_2$ based on the Michelson–Levy birefringence chart) using a CI-S POL optical microscope (Nikon, Japan) a ×10 objective and DS-Rs1 camera (Nikon, Japan). The thickness of the sample is measured in the microscope image, the interference colour is located in the first and second order. Segments were immersed in deionized water for 20 s before post drawing in the air using the same Autograph universal tester with a constant drawing speed of ~5 mm/min until an increase of modulus was seen, where at this point drawing is stopped and the current stress on sample maintained for 5 min until the sample is dried.

Mechanical properties were determined based on 5 sample repeats per progression stage using the same Autograph universal tester (Shimadzu, Japan) equipped with a 5 N load cell and a strain rate of 2 mm/min. The time between each measured stress data point was set to 0.05 s. Samples had an initial gauge length of 4 mm and were gripped on an universal clamp using abrasive paper and adhesive tape.

As the strain accumulation in a drawn sample happens in only a few millimetres, samples with calculated different accumulated extensional strains in Figs. 3 and 4 are not segments of a single drawn sample. They are prepared by individual drawing process, respectively, and their strain and properties are measured after drawing.

A Rosand RH10 (Malvern Panalytical, UK) capillary rheometer was applied in the extrusion process. The barrel for the input solution is 15 mm in diameter and the exit mold is 0.5 mm in diameter and 32 mm in length. The shear rate was controlled to achieve different accumulated extensional strain and the pushing speed of the piston in the barrel was 4 mm/min.

The flow field of both drawing and extrusion spinning is simulated using a COMSOL software 5.4_with measured parameters input parameters. The transient speed field of the aqueous protein system is converted into accumulated extensional strain according to the calculations, assumptions and approximations found in the Supplementary Information. For input into flow simulation, silk solution rheological properties were measured with a MCR 302 (Anton Paar, Austria) rotational rheometer with a Peltier controlled bottom plate at 25 °C with a 25 mm diameter 1-degree cone and plate at 0.09 mm truncation gap geometry gap. All measurements were well within the torque limit for accurate measurement.

**Sample characterisation, imaging and Raman spectroscopy measurements.** SEM images were taken using a Nova 450 microscope (FEI, ThermoFisher, USA), the fibre/solid samples were imaged using 1 keV incident electron energy without applying a conductive coating. BSE images were collected by a concentric backscattered detector to maximise the material contrast. All cryo-cross sections were prepared by a direct fracture in liquid nitrogen, and longitudinal section samples were prepared by fracturing the sample in liquid nitrogen at an angle. In some situation, the top of the sample partially breaks off, revealing the longitudinal section. The non-cryo samples are prepared by a simple stress fracture.

Raman spectra of samples were measured using a DXRxi Raman spectrometer (ThermoFisher, USA) using a 532 nm <5 mW power laser, where the signal to noise ratio in 600–2000 cm⁻¹ wavenumber region is above 8.0.

The beta-sheet fraction of the sample can be calculated by measuring the crystalline phase occupation in BSE cross-sectional images or calculating the peak ratio between the 1665 and 1400 cm⁻¹ bands in Raman spectra. The water-dissolvable fraction of samples are measured as follows: Samples was stored in water for one day at room temperature to remove the dissolvable content. The remaining solid was dried and weighed, and is considered to be only non-dissolvable protein. The pocket volume fraction is calculated through measuring pocket occupation in BSE cross-sectional images using Image J 1.52a (NIH Software, USA). Water expelled is calculated by measuring the weight loss after carefully removing any water droplets that appeared on the sample surface.

## Data availability
The authors declare that the main data supporting the findings of this study are available within the article and its Supplementary Information files. Extra data are available from the corresponding author upon request.

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

## Acknowledgements

We thank The Royal Society and National Natural Science Foundation of China for funding through the Royal Society-Newton Mobility Grant (IEC\NFSC\181308 and NFSC-81911530223). M.Y. thanks for support by the following funding bodies: NSFC (31800807), Zhejiang provincial natural science foundation of China (LZ17C170002), State of Sericulture Industry Technology System (CARS-18-ZJ0501), Fundamental Research Funds for the Central Universities (2018XZZX001-11), Zhejiang Provincial Science and Technology Plans (2016C02054-19) and Zhejiang Provincial Key Laboratory of Utilisation and Innovation of Silkworm and Bee Resources (2020E10025). C.R. thanks EPSRC for support under EP/N008065/1 and C.H. thanks EPSRC for support under EP/K005693/1. Y.S. thanks for support by Fundamental Research Funds for the Central Universities (2020QNA6028). We would also like to thank Dr Dandan Song (CCEM Zhejiang University), Dr Jing He (State Key Laboratory of Chemical Engineering of Zhejiang University), Dr Yan Liu (School of earth science experimental center of Zhejiang University), College of Biosystem and food science experimental center of Zhejiang University and Dr Rui Mi (Liaoning Academy of Agricultural Sciences) for help with experiments, and Prof. Fritz Vollrath (Oxford University), Prof. Mao (Oklahoma University), Prof. Guan (Beihang University) and Dr Dicko (Lund University) for insightful pre-submission manuscript review.

## Author contributions

Q.W. designed the experiments, performed the laboratory work, analysed data and drafted the manuscript; Y.S. participated in Raman spectra acquisition and analysis; Mei Yang, J.H., F.L. and J.W. participated in laboratory work; C.H., C.R. and M. Y. directed the work and revised the manuscript.

## Competing interests

The authors declare no competing interests.
