## [Peer Review File · Nature Communications]

Reviewers' Comments:

Reviewer #1:

Remarks to the Author:

Wan et al. revised the manuscript according to the reviewers' comments. The reviewer 1 is almost convinced by their revisions, but still have two concerns as follows:

"Q1: 1. In the manuscript, they do not distinguish spider silk and silkworm silk very much. They might consider both hierarchical structures and assembly mechanisms are similar or same. However, the reviewer considers the molecular mechanism of spider silk (especially, dragline MaSp) is different from that of silk worm silk like *B. mori* silk. To clear this confusion, it might be better to clear the topic in this manuscript is about silkworm silk. Otherwise, it looks confusing both types of silk fibroins.

We apologise for the confusion this lack of distinction, whilst the presence of mesoscale structures has been found in a range of silks from both spiders and silkworms. We have amended the manuscript to account for this diversity and the fact that our results focus on silk proteins derived from *B. mori* silk. Lines 30-33 and 45-47"

New comment

They amended to account for the species diversity, namely, silkworm silk. However, the title and abstract still do not contain any species explanation. The reviewer recommend to include "silkworm silk" in the title.

"Q3: As shown in Figure 1, the silk fiber prepared in this study showed porous region and also non-round shape morphology. These shapes (porous and non-round) are different from the natural silk and also seem to affect the share force distributions. The mechanisms and simulation's setting for Figure 2, 3, and 4 are really matching to the porous and non-round shape?? It could be nice to add a reasonable explanation for this point.

We thank the reviewer this observation as it has provided us with the opportunity to provide further clarification in the manuscript. For the simulations and mechanisms proposed we assumed that the deformation applied was acting on a droplet and thus a circular cross sectional area, with any deviations from this a consequence of dehydration prior to imaging. The fibre shown in figure 1 is a point in fact, as it was chosen specifically because it demonstrated a variety of meso-scale features, however the non-round overall shape can be accounted for as this section for it is nearly 20 times bigger than a natural silk fibre and thus we assume some dehydration has caused a change in shape. As a result we have changed the manuscript line 56-61."

line 56-61. "This cross section of a partially spun regenerated silk fibre (Fig. 1a) clearly contains features that have been seen in natural silkworm silk fibres that are an order of magnitude smaller in diameter (Fig. 1b-e), such as fibrillar structures on the surface¹³ and the presence of nanoscale voids which we propose are more appropriately referred to as "pockets"³². Notably to our knowledge these pockets are not readily observed in *B. mori* silks, but are in wild silkworm silks, suggesting that they may be under different selective pressures (artificial vs. natural)."

New comment

The reviewer understood the additional explanation (line 56-61). If the fiber diameter and morphology are very different between silkworm silk and the regenerated silk fiber (Figure 1), the pocket structures (nano-scale voids) are so critical for the physical properties and also spinning? Also, *B. mori* silk fiber hardly shows these pockets, but wild silkworm silk does, even though the spinning systems of *B. mori* and wild silkworm are similar. These results imply the pockets in silk are not so reproducible and essential. Furthermore, Figure 1b, c, d, and e are not so clear enough to compare with Figure 1a. It might be nice if they reorganize Figure 1 to compare those figures clearly.

Reviewer #3:

Remarks to the Author:

I have reviewed the publication and feel the authors have done a solid job in responding to the prior critiques and I remain very positive about the contribution. I only have one major point that still needs addressing, and a few minor points.

Major: The major point that remains unclear, which also seems to be a key to entire paper, is the starting regenerated silk material. I appreciate the inclusion of some of the data on molecular weight as well as the SDS-PAGE gel in the revision, but this raises more questions. A key in the paper according to the authors is that the method of silk isolation is more gentle/different, so that the process preserves more native structures/molecular weights (e.g., "...our regenerated protein is much closer to natural silk..."). Yet, the materials used had a significantly lower molecular weight than the native material (~400 kDa native vs. 120 kDa for the material generated in the present work), so I fail to see how the isolation method is any less harsh than other isolation methods in other papers on regenerated silk solutions, albeit obtained from different process conditions. If the claim is that chain scission is less, this does not seem to be supported by the data provided. Also, the other claims ascribing lower molecular weight fractions to sericins, light chain or P30 seem to be without analytical support, as size alone is not sufficient here without appropriate controls, amino acid analysis of fractions or related techniques to determine content. I think it is important to clarify the above points to make sure the work and claims are reflective of the data. Further, if the above is correct, then the authors need to try and clarify the origins of the structures observed; for example, are these due to residual light chain, P30 or other factors, in terms of the various conclusions in the paper.

Minor: Figure 1 is much improved, thank you. Please just add a notation for the yellow arrows in the figure legend so this is clear to the reader what these are pointing to.

Minor: p9, line 291-292 – is this correct, as the references cited do not seem to match the prior question on what happens to all of this water. With 70% water, that is a lot to lose so quickly upon exposure to air, particularly in humid environments. Please clarify if possible.

Response to reviewers' comments

We thank the reviewers for their thoughtful suggestions and opinions, it was a great set of peer reviews and we thank them for their time and expertise. We have endeavoured to address their comments in full, all of which have helped in strengthening the clarity, breadth and impact of the work presented. To simply the assessment of our response **we've used red text to mark both our response and changes made to the manuscript.**

Reviewer #1 (Remarks to the Author):

Wan et al. revised the manuscript according to the reviewers' comments. The reviewer 1 is almost convinced by their revisions, but still have two concerns as follows:

"Q1: 1. In the manuscript, they do not distinguish spider silk and silkworm silk very much. They might consider both hierarchical structures and assembly mechanisms are similar or same. However, the reviewer considers the molecular mechanism of spider silk (especially, dragline MaSp) is different from that of silk worm silk like *B. mori* silk. To clear this confusion, it might be better to clear the topic in this manuscript is about silkworm silk. Otherwise, it looks confusing both types of silk fibroins.

We apologise for the confusion this lack of distinction, whilst the presence of mesoscale structures has been found in a range of silks from both spiders and silkworms. We have amended the manuscript to account for this diversity and the fact that our results focus on silk proteins derived from *B. mori* silk. Lines 30-33 and 45-47"

New comment

They amended to account for the species diversity, namely, silkworm silk. However, the title and abstract still do not contain any species explanation. The reviewer recommend to include "silkworm silk" in the title.

We apologize for not being clear enough in title and abstract. We agree with the reviewer and have included term "silkworm silk" in the title and species in the abstract accordingly to avoid any confusion with different silk spinning animals.

"Q3: As shown in Figure 1, the silk fiber prepared in this study showed porous region and also non-round shape morphology. These shapes (porous and non-round) are different from the natural silk and also seem to affect the share force distributions. The mechanisms and simulation's setting for Figure 2, 3, and 4 are really matching to the porous and non-round shape?? It could be nice to add a reasonable explanation for this point.

We thank the reviewer this observation as it has provided us with the opportunity to provide further clarification in the manuscript. For the simulations and mechanisms proposed we assumed that the deformation applied was acting on a droplet and thus a circular cross sectional area, with any deviations from this a consequence of dehydration prior to imaging. The fibre shown in figure 1 is a point in fact, as it was chosen specifically because it demonstrated a variety of meso-scale features, however the non-

round overall shape can be accounted for as this section for it is nearly 20 times bigger than a natural silk fibre and thus we assume some dehydration has caused a change in shape. As a result we have changed the manuscript line 56-61."

line 56-61. "This cross section of a partially spun regenerated silk fibre (Fig. 1a) clearly contains features that have been seen in natural silkworm silk fibres that are an order of magnitude smaller in diameter (Fig. 1b-e), such as fibrillar structures on the surface¹³ and the presence of nanoscale voids which we propose are more appropriately referred to as "pockets"³². Notably to our knowledge these pockets are not readily observed in *B. mori* silks, but are in wild silkworm silks, suggesting that they may be under different selective pressures (artificial vs. natural)."

New comment

The reviewer understood the additional explanation (line 56-61). If the fiber diameter and morphology are very different between silkworm silk and the regenerated silk fiber (Figure 1), the pocket structures (nano-scale voids) are so critical for the physical properties and also spinning? Also, *B. mori* silk fiber hardly shows these pockets, but wild silkworm silk does, even though the spinning systems of *B. mori* and wild silkworm are similar. These results imply the pockets in silk are not so reproducible and essential. Furthermore, Figure 1b, c, d, and e are not so clear enough to compare with Figure 1a. It might be nice if they reorganize Figure 1 to compare those figures clearly. (depends on exact spinning condition and we present a few example for different condition and resulting fibers?)

We apologize for lack of discussion around this area which we have elaborated further through inclusion of more references surrounding void-like structures in silk fibres, and identified areas for future research in the revision (page 2 line 56 and 64 and Pages 3 line 66). Specifically we have highlighted the transitory nature of these features and as a result may or may not be present depending on species, spinning conditions and sample preparation.

Reviewer #3 (Remarks to the Author):

I have reviewed the publication and feel the authors have done a solid job in responding to the prior critiques and I remain very positive about the contribution. I only have one major point that still needs addressing, and a few minor points.

Major: The major point that remains unclear, which also seems to be a key to entire paper, is the starting regenerated silk material. I appreciate the inclusion of some of the data on molecular weight as well as the SDS-PAGE gel in the revision, but this raises more questions. A key in the paper according to the authors is that the method of silk isolation is more gentle/different, so that the process preserves more native structures/molecular weights (e.g., "...our regenerated protein is much closer to natural silk..."). Yet, the materials used had a significantly lower molecular weight than the native material (~400 kDa native vs. 120 kDa for the material generated in the present work), so I fail to see how the isolation method is any less harsh than other isolation methods in other papers on regenerated silk solutions, albeit obtained from different process conditions. If the claim is that chain scission is less, this does not seem to be supported by the data provided.

Also, the other claims ascribing lower molecular weight fractions to sericins, light chain or P30 seem to be without analytical support, as size alone is not sufficient here without appropriate controls, amino acid analysis of fractions or related techniques to determine content. I think it is important to clarify the above points to make sure the work and claims are reflective of the data. Further, if the above is correct, then the authors need to try and clarify the origins of the structures observed; for example, are these due to residual light chain, P30 or other factors, in terms of the various conclusions in the paper. (our focus on structure not molecule, additional data in SI)

We apologise for the continued lack of clarity here, we agree that the use of a regenerated silk will undoubtedly introduce variation that is not present in the natural system. However for the purposes of this work, our method for resolubilization produced a more “native-like” material measured not specifically through maintenance of protein chain length, but through maximizing solution viscosity, data which is now included in the supporting information. The higher viscosity observed relative to other regenerated silk protocols suggests that chains were sufficiently long and entangled to maintain viscosity (despite scission clearly happening as we see in the accompanying gels). Regarding the commentary about the lower MW components, we agree with the reviewer, this was overstretching our interpretation of the data and as a result we have removed this discussion from the supporting information.

Minor: Figure 1 is much improved, thank you. Please just add a notation for the yellow arrows in the figure legend so this is clear to the reader what these are pointing to.

We thank the reviewer for kind suggestion and we have amended our figure 1 description (page 2 line 54) accordingly.

Minor: p9, line 291-292 – is this correct, as the references cited do not seem to match the prior question on what happens to all of this water. With 70% water, that is a lot to lose so quickly upon exposure to air, particularly in humid environments. Please clarify if possible.

We apologize for not clear enough with this point. We have now introduced the influence of environmental humidity on fibre dehydration and formation as supported by recent studies in this area. page 4 line 125.

As a final note, we would really like to thank the reviewers for finding the time to spend on offering their thoughts for our manuscript, with their help we hope that it will be of use to the wider silk community and hopefully spark ideas and avenues for further research.

Reviewers' Comments:

Reviewer #1:

Remarks to the Author:

Thank you for your nice revisions. I'm convinced and happy to agree the publication.

Response to reviewers' comments

Reviewer #1 (Remarks to the Author):

Thank you for your nice revisions. I'm convinced and happy to agree the publication.

We would like to thank the reviewer for their time and expertise in previous thoughtful revisions.